# Reconceptualising the Digital Gender Divide, Accommodating New Forms of Virtual Gender-Based Violence

**DOI:** 10.3390/bs15111568

**Published:** 2025-11-17

**Authors:** Elena López-de-Arana Prado

**Affiliations:** Departamento de Pedagogía, Facultad de Formación de Profesorado y Educación, Universidad Autónoma de Madrid—Campus de Cantoblanco, Calle Francisco Tomás y Valiente, n° 3, 28049 Madrid, Spain; elena.lopezdearana@uam.es; Tel.: +34-91-497-6036

**Keywords:** gender digital gap, digital gender-based violence, childhood, adolescence, critical digital literacy, health and well-being, feminism

## Abstract

From a critical feminist perspective, it is hypothesised that the gender digital divide may be related to new forms of virtual gender-based violence that particularly affect girls and young women. If this is the case, these forms of violence would fall within the dimension of exploitation or quality of use of technologies that characterises the digital divide. To test this hypothesis, a documentary analysis of the phenomenon was carried out by reviewing different cases reported in various media outlets, which show that the well-being of girls and adolescents is at risk when technology is involved. Four categories emerge that reflect situations in which technology becomes a tool for promoting self-harm and suicide among minors through exposure to harmful content, grooming, sexting and/or sextortion; the digital sexual exploitation of underage girls through deepfakes or intimate images generated with artificial intelligence; the consumption of violent and hateful content in mass chats; and the incitement of gender-based violence through video games. The results show the reproduction and perpetuation of gender-based violence in the digital world. To guarantee safe, inclusive and equitable digital environments, various measures are essential, including European policies or plans aimed at guaranteeing digital security and rights, and those related to critical digital literacy with a gender perspective in formal education (school and university) and informal education (parents, carers and guardians). Finally, we urge that the focus be placed on personal digital resilience, since thinking of a completely secure digital world is a naive and unattainable utopia.

## 1. Introduction

The rapid digitisation of everyday life has not only transformed the way people communicate, learn and interact, but has also intensified existing social inequalities, particularly those based on gender ([56]). In this context, understanding the digital gender divide requires a multidimensional perspective that goes beyond access to technology and encompasses disparities in skills, participation and the ability to use digital resources for personal, educational and civic empowerment purposes ([59]).

This paper examines the intersection between the digital gender divide and gender-based digital violence, highlighting how certain dynamics in digital environments expose girls and young women to specific risks that threaten their well-being and rights.

The theoretical framework is structured around three interrelated dimensions. First, Digital Competence within the European DigComp Framework is analysed, which offers a comprehensive model for assessing and promoting digital competences and critical engagement. Second, the *Dimensions of the Digital Divide*—access, use and outcomes—are reviewed to reveal the persistence of structural inequalities in digital inclusion. Finally, the *Digital Gender Divide According to Dimensions* is analysed, illustrating how certain patterns of interaction amplify the exposure of girls and young women to digital gender-based violence and hinder the creation of a safer and more inclusive digital ecosystem environments.

## 2. Digital Competence Within the European DigComp Framework

Contemporary society is undergoing a profound digital transformation, which has elevated digital education to the centre of educational strategies. In this context, digital competence is recognised as one of the basic skills for lifelong learning and active participation in an increasingly digitalised world.

The European Digital Competence Framework for Citizens (DigComp) emerged as the main regulatory benchmark promoted by the [20] ([20]). DigComp defines digital competence as the safe, critical, and responsible use of digital technologies for learning, work, and participation in society, including effective interaction with these tools ([63]).

The most recent version, DigComp 2.2, continues to structure this competence into five key areas ([63]): (1) information and data literacy, (2) communication and collaboration, (3) digital content creation, (4) safety (including digital well-being and cybersecurity-related competences), and (5) problem solving. These areas are broken down into 21 specific competences with eight levels of mastery (foundation encompassing levels 1 and 2, intermediate encompassing levels 3 and 4, advanced encompassing levels 5 and 6, and highly specialised encompassing levels 7 and 8), providing detailed guidance on what every citizen should know, know how to do, and be in digital environments (see Figure 1).

The most relevant additions to DigComp 2.2 are detailed below. On the one hand, in line with emerging concerns, special attention is paid to the identification and management of misinformation and disinformation on social media and news sites ([38]; [63]). On the other hand, it also includes the management of what are known as “advanced technologies” related to robotics, artificial intelligence (AI), extended reality (augmented and virtual), and the Internet of Things (IoT) ([63]). These considerations are further enhanced by the incorporation of risks associated with “datafication” and environmental sustainability ([63]). Finally, specifically within the area of security, the sub-competence “4.3. Protection of health and well-being” stands out, revealing a direct link between the framework and digital well-being ([63]).

Although the DigComp 2.2 framework does not explicitly address the contextual variables that influence digital competence, numerous studies indicate that access to technology, as well as skills in its use and exploitation, are conditioned by factors such as age, gender, place of residence, race, educational level, and income ([24]; [61]). The differences or inequalities in access to, use of, and utilisation of ICTs among different social groups, territories, or individuals are referred to as the digital divide, which is yet another manifestation of the structural social inequalities that surround us ([56]).

## 3. Dimensions of the Digital Divide

The concept of the digital divide has evolved significantly since its inception in the late 20th century to the present day. Initially, it was simply defined as the difference between those who had access to the Internet and Information and Communication Technologies (ICTs) and those who did not ([24], [25]; [61]). This quantitative conception focused mainly on connectivity. However, with the growing complexity and ubiquity of the digital revolution, the limitations of this definition have become apparent, revealing its reductionist nature. Currently, the digital divide is understood to reflect the existing “enormous social divide,” which encompasses a wide range of inequalities that go beyond simple access ([61]; [47]).

Currently, three main dimensions of the digital divide are recognised: the access divide, the usage divide, and the quality of use or the empowerment divide ([45]; [47]).

The first dimension encompasses the access divide, which includes equipment and connectivity. This divide refers to the disparity in people’s ability to access the Internet and ICTs ([45]; [47]). It manifests as a lack of access to the necessary digital devices, software, or a network connection ([45]; [47]). Although this access divide has been significantly reduced worldwide, it is worth noting that it has not been completely eliminated. In countries such as Spain, for example, in 2023, 8.4% of the Spanish population still lacked internet access ([25]). This divide is accentuated by sociodemographic factors, including age, gender, race, educational level, and socioeconomic or income level ([61]).

The digital usage or skills divide constitutes the second dimension and is related to differences in the level of knowledge and skills needed to function adequately in the digital world ([45]; [47]). It is not enough to have access—to have a device and a network—it is essential to understand what the Internet is, what social networks are, what fake news is, what artificial intelligence is, and how they work, as well as what can be carried out and how to carry it out. Therefore, this divide is related to the acquisition of digital competence, which is facilitated by the education system ([36]; [47]). It primarily affects older individuals, those who have not completed compulsory education, women, and individuals on low incomes ([24]; [36]; [45]).

The third and final dimension is the divide in the quality of use or empowerment. This divide focuses on how access to and use of technologies translate into concrete and effective benefits for all people, without exception ([45]). This divide arises when people lack equal opportunities to utilise ICTs and when there is a lack of digital empowerment equality. Currently, this divide has attracted greater interest because the skills divide tends to overlook the importance of using technologies in a meaningful, healthy, responsible, ethical, critical, and safe manner, ignoring digital inclusion and the violence that can emerge in this new digital age. However, it is still in the process of being properly conceptualised with all its vicissitudes. It is important to emphasise that it is linked to the cultural and relational capital of the individual’s environment, including the family context and, above all, the school context ([47]). Digital education must therefore transcend the mere teaching of instrumental skills to incorporate support and the development of critical thinking, ensuring that access and use translate into real empowerment; the absence of any type of exclusion—which is often related to different functional diversities—or violence—which is often intersected by the component of gender diversity or affective-sexual orientation; and the attainment of tangible benefits (social, cultural, economic, personal, political, etc.) ([45]).

## 4. Digital Gender Divide According to Dimensions

As detailed in the previous section, gender, that is, being a woman or not conforming to heteronormative gender stereotypes, makes certain groups digitally vulnerable ([24]; [36]; [45]; [61]). However, despite this evidence, the digital gender divide is one of the most persistent and often underestimated inequalities.

Women’s inequality in access to devices and/or connectivity has been the most prominent dimension of the digital gender divide. For more than 10 years, various individuals have focused on this issue ([61]). However, more recent analyses indicate that the digital gender divide persists worldwide, with 70% of men using the Internet, compared to 65% of women. Furthermore, the number of women who are disconnected is 17% higher than that of men ([28]).

The digital gender divide is not only a question of access, but also of use and appropriation of technology ([61]). Historically, technology has been designed “by and for men,” which has detracted from women’s prominence and interest in its use, as well as the development of their digital skills ([61]; [48]). Masculinised designs and the lack of female role models in scientific and technological fields may be contributing to girls’ lack of motivation in STEM (science, technology, engineering, and mathematics) studies. Consequently, in ICT-related professions, women represent only 1.7% of ICT specialists in total employment, compared to 6.2% for men; there is also a gender pay divide of 9% ([25]). This trend is particularly curious, as girls often achieve better aptitude results in STEM areas, yet paradoxically, their presence ends up being scarce ([25]). Therefore, given this reality, it can be said that the stereotypes of “female technophobia” and “male technophilia” discussed by [61] ([61]) persist today.

Finally, in the digital gender divide, the dimension related to the quality of use or the empowerment is intertwined with the emergence of exclusion, traditionally associated with agency and autonomy. However, this dimension could also reveal the persistence of systemic gender inequalities.

To contextualise this issue, it must be recognised that the binary system imposed by our society reproduces stereotypes and roles, leading to exclusion and, consequently, suffering for all those who do not comply with heteronormative mandates related to gender identity and/or affective-sexual orientation ([26]). The absence of feminist or queer pedagogies is the perfect breeding ground for ensuring the continuity of sexism, machismo, homophobia, and transphobia ([26]). If we add to this uncritical digital literacy, girls, young women, and, in general, all those who do not conform to heteronormative mandates become potential targets of virtual violence ([29]; [44]; [52]; [57]). Among the various manifestations of digital violence, cyberbullying stands out, and specifically, sexist cyberviolence, which manifests itself when threats or attacks are directed against women and girls with an explicit component of domination, control, humiliation, or sexual exploitation. Examples of these practices include harmful content and hate speech shared on social media or in mass chats, and gender-based violence in immersive environments such as video games, sexting, sexual deepfakes, grooming, sextortion, and revenge porn (see Table 1).

Therefore, without a critical gender perspective in digital literacy, young people are bound to reproduce stereotypes in a more radicalised way in the virtual world, to normalise any form of violence, and specifically gender-based violence, which will jeopardise their own well-being and that of others, both now and in the future.

With this in mind, the aim of this paper is to examine the relationship between the empowerment dimension of the digital gender divide and the digital gender-based violence, identifying its manifestations and consequences on the well-being of minors, in order to put forward proposals that will enable the creation of a safer and more inclusive digital environment.

## 5. Documentary Analysis as a Method for Gathering News About Digital Gender-Based Violence

A documentary analysis was carried out of journalistic sources that report cases of gender-based cyberviolence suffered by minors. The documentary analysis is based on the methodological proposal of [6] ([6]), who conceives of documents as significant sources for understanding social and cultural phenomena.

### 5.1. Selection of the Research Corpus

The review of journalistic sources covers the period from 2012, when one of the first high-profile cases was reported, to the present day, 2025. Therefore, the corpus analysed covers thirteen years of media coverage.

In this study, the criterion for including cases reported in the news was their high visibility or wide media coverage. To this end, cases had to meet the following requirement: they had to be published in at least 20 national or international generalist media outlets.

### 5.2. Categorisation of Cases According to Their Impact on Child Well-Being

Cases were categorised according to their impact on child well-being, understood as the emotional, psychological and social effects of different forms of gender-based cyberviolence on minors. This approach made it possible to organise cases in a coherent manner, prioritising the observed consequences over the type of platform or technological mechanism used.

### 5.3. Analysis Procedure

The procedure was structured in three complementary phases, following the classic guidelines for qualitative studies of document analysis based on triangulation of sources ([6]; [14]; [40]).

#### 5.3.1. Phase 1: Identification and Selection of the Corpus

To ensure that the selection of cases was systematic, transparent, and replicable, the following steps were followed:-Establishment of criteria for inclusion of cases (minimum number of media outlets reporting the case);-Compilation of a list of cases in which underage girls had been victims of sexist cyberbullying during the years 2012–2025;-Review of Google News—where the data is observable—to determine whether the cases met the established criteria—not all news items that appear are counted, but rather it is checked whether they actually deal with the case, and some of them must be removed;-Use of different Generative Artificial Intelligences (ChatGPT (GPT-5.1 model), Copilot (GPT-5 model), etc.) to increase the list of cases and review their presence on Google News again.

#### 5.3.2. Phase 2: Categorisation of the Corpus

Following [30] ([30]), a thematic analysis of the content of the news items is carried out, taking into account the type of gender-based violence and the degree to which it affects the well-being of minors.

Four main categories emerge (see Table 2): (1) self-harm and suicide among minors with two subcategories that explain the type of violence that leads minors to take their own lives: grooming, sexting or sextortion, and exposure to harmful content; (2) digital sexual exploitation of underage girls through deepfakes; (3) consumption of violent and hateful content in mass chats; (4) incitement to gender-based sexual violence through video games.

#### 5.3.3. Phase 3: Triangulation of Sources

To ensure the validity and interpretative completeness of the study, different sources are triangulated ([14]): information obtained from the media, scientific literature, institutional reports, and judicial documents.

## 6. Emerging Results: When Digital Technology Is “Exploited” Against Girls: New Forms of Gender-Based Violence That Threaten Their Well-Being

The analysis reveals one of the dark sides of the digital age, that is, the alternative forms of gender-based violence that manifest and spread through instant messaging, social media, and video games. The news of the cases analysed highlight how technology can become a tool for promoting self-harm and suicide among minors through exposure to harmful content, grooming, sexting, and/or sextortion; the digital sexual exploitation of underage girls through deepfakes or intimate images generated with AI; the consumption of violent and hateful content in mass chats; and the incitement of gender-based sexual violence through video games.

Below are several cases that formed part of the documentary analysis and illustrate these issues.

### 6.1. Self-Harm and Suicide Among Minors

If we review the news stories that point to the digital world as responsible for encouraging and inducing young people to engage in self-harming behaviour or suicide attempts, we find a significant number. There are cases where the impact is related to exposure to certain content that incites self-harm and suicide, but there are other cases where the outcome is death because adults are extorting them.

#### 6.1.1. Due to Grooming, Sexting, and Sextortion

When the outcome of a digital interaction is that a minor self-harms or takes their own life, we are dealing with cases of grooming, sexting, and sextortion. Grooming and sexting are the crimes that precede sextortion. Grooming occurs when an adult tries to gain the trust of a minor to manipulate and sexually exploit them. Sexting often occurs in this interaction, as the adult usually asks the minor for sexual photos or videos at some point. These practices sometimes evolve into real-life encounters or sextortion, threatening the minor with publishing the content if they do not comply with the requests.

The case of Amanda Todd was a tragic example of the consequences of sextortion. [41] ([41]) recounted the ordeal of Amanda, who at the age of 15 ended up committing suicide in Canada after suffering cyberbullying since the age of 12. At that age, a stranger contacted her online and asked her for a photo of her breasts. She agreed, and from then on, her life became hell. The cyberbully blackmailed her into performing an “online dance”, which apparently involved a striptease. When Amanda refused, the attacker carried out his threat and published her topless photo, even creating a Facebook profile with the photo of her bare breasts as the profile picture. Although she moved and changed schools, the image always reappeared in her new environment. This image motivated her classmates to bully her. Self-harm and suicide attempts alternated with virtual and real attacks. Finally, Amanda took her own life, but days before, she recorded a video in which she recounted her torment. This devastating case of sextortion leading to bullying triggered awareness and the emergence of considering these situations within the legal framework.

Another of the most alarming cases came to light in April 2025, when the US Department of Justice announced the dismantling of the international child exploitation network known as “764”. The network operated with a high degree of technological sophistication ([58]). The leaders of the network were charged with multiple crimes, including the production and distribution of child sexual abuse material, grooming of minors, and various forms of sextortion. Between 2020 and 2025, at least eight teenagers—some as young as thirteen—were direct victims, forced to self-harm, participate in sexual acts both online and in person, abuse animals, and even assault family members, in a pattern of planned and repeated violence. The prosecution sought life imprisonment for those responsible.

More than a decade has passed between the cases presented here. The case of Amanda Todd represents the social awareness of the tragic realities that can be experienced on social media. And the dismantling of the “764” network highlights the evolution of grooming and sextortion. Whereas victims previously faced a single cyberbully, they are now or may be extorted by technologically sophisticated organisations with patterns of cyberviolence shared by a large, more structured community. The impact on victims remains devastating, as it always leads to profound deterioration in mental health, depersonalisation of minors, and, in extreme situations, self-harm and suicide. What has changed substantially is the difficulty in detecting cyberbullies, their capacity for expansion, and their resistance to disappearing.

Given this reality, critical digital literacy and awareness of children’s digital rights are crucial, as they equip them with the tools to safeguard their safety ([44]; [57]). When self-protection fails, awareness of the possibility of reporting and doing so without delay is key to dismantling cybercriminal networks that engage in digital violence. And if these are international, the establishment of global legal frameworks, such as the European one, will be of great help ([19]).

#### 6.1.2. Due to Exposure to Harmful Content

Responsibility in dealing with this issue is essential because not all cases reported in the press are true or have clear evidence to support them.

In this section, we will look at two cases that were widely reported in the media at the time, but for which there was never any evidence linking them to self-harming behaviour or suicide attempts among young people. These are the “Blue Whale Challenge” and the “Momo Challenge”.

The first, the Blue Whale Challenge, began in Russia in 2015, and in 2016 rumours began to spread about the existence of an alleged online “suicide game”. The challenge, according to rumours, consisted of completing 50 tasks in 50 days, ranging from watching horror films to self-harm, culminating in a final challenge: taking one’s own life ([2]). Alleged suicides due to this game were reported in Russia, the United States, and India ([2]). However, subsequent investigations indicated that the “Blue Whale Challenge” may not have existed in a concrete form until its existence was published and disseminated in the traditional media ([2]).

The second game, the Momo Challenge, was another alleged viral phenomenon that originated in Japan and appeared to have spread worldwide via WhatsApp, Facebook, and YouTube. It was described as a macabre game that culminated in the death of the participants. Despite reports of alleged cases in different parts of the world: Germany, Argentina, Belgium, Colombia, Scotland, the United States, France, and India ([1]; [27]), investigations found no evidence to link suicides among young people with participation in the game ([64]).

In addition to the impact of the “Blue Whale Challenge” and “Momo Challenge” games on self-harming and/or suicidal behaviour in young people not being considered accurate, it is noteworthy that none of the cases are good examples of the differential impact that the digital world has across genders. The false cases do not confirm these issues. This is noteworthy and significant because the true cases do, such as the example of Molly Russell, which shows that girls are more vulnerable to certain content and may suffer more from anxiety, depression, etc. ([53]).

The case of Molly Russell illustrates how social media algorithms can promote self-harm and suicide, particularly among girls and adolescents. According to [65] ([65]), in 2017, Molly, a 14-year-old from London, took her own life after consuming a massive and repeated amount of content related to depression, self-harm, and suicide on Instagram and Pinterest. Her father, Ian Russell, described the content as “the darkest of worlds,” an “online ghetto” where the algorithm prevents escape and continues to recommend more similar content ([65]). During the court proceedings, Elizabeth Lagone, a representative of Meta, admitted that the content viewed by Molly was “completely inappropriate” by the platform’s standards or expected “regulation” ([65]). Finally, the UK court ruled that Molly died as a result of “an act of self-harm while suffering from depression and the negative effects of online content,” which was an explicit acknowledgement of the responsibility of algorithms in this outcome ([65]).

The tragic case of Molly Russell highlights the real risk that access to content related to self-harm and suicide on social media poses to the well-being of girls and young women. However, it should be noted that the impact of this type of content in digital environments is not as widespread as has sometimes been claimed; that is, not all—or even the majority—of young people who access it end up self-harming or committing suicide. It would therefore be interesting to reflect on what interest might lie behind the social construction of this belief, which leads us to demonise ICTs ([23]; [31]; [34]), and to jeopardise the possibility of critical digital literacy, especially when the institutional response translates into the elimination or prohibition of technology in educational centres ([8]; [44]; [57]).

### 6.2. Digital Sexual Exploitation of Underage Girls Through Deepfakes

An alarming manifestation of digital gender-based violence is the creation and dissemination of sexual images of minors using artificial intelligence (AI). In September 2023, dozens of underage girls in Almendralejo, Extremadura (Spain), accompanied by their mothers, reported the creation of images manipulated with artificial intelligence in which fake nude images of them appeared and their subsequent dissemination via mobile devices ([62]). The judicial police identified several alleged perpetrators and referred the case to the Juvenile Prosecutor’s Office ([9]). However, the main problem was the legal vacuum that existed, and still exists, in Spain, as there is no specific law regulating AI-generated pornography or the digital sexual exploitation of underage girls through deepfakes ([33]). Finally, 15 schoolchildren were found guilty of 20 counts of child pornography and 20 counts of crimes against moral integrity, and the sentence included one year of probation, with affective–sexual education, gender equality awareness, and responsible use of ICTs ([9]).

These cases clearly highlight the differences in the use of ICTs across genders, as the literature shows that children and young people tend to be more prone to engaging in risky online behaviour ([53]), in these cases, digital sexual exploitation. However, what is not usually pointed out is that these risky or violent behaviours are often directed towards girls and/or women, and the focus is usually on the emotional–sexual level. Aware of these disparities, we can therefore consider the need for personalised interventions that take into account the diverse experiences and needs of different genders. The ruling on the Almendralejo children highlights potential avenues for educational diversification, focusing on effective sexual education, gender equality awareness, and the responsible use of ICTs for boys ([9]). To this proposal, we could add the promotion of empathy—both cognitive and affective—moral commitment, and the introjection of norms ([12]; [15]). On the other hand, without forgetting the female gender, in the case of women, empowerment should be facilitated through sisterhood ([32]). This trend emerges spontaneously and naturally, as seen in the case of Spain, where mothers provide unconditional support to their daughters, resulting in a support network among them.

### 6.3. Consumption of Violent and Hateful Content in Mass Chats

The proliferation of mass chats on instant messaging platforms such as WhatsApp, where minors are introduced without their consent and inappropriate and criminal content is disseminated, is another worrying manifestation of digital violence. In different parts of Spain—Cantabria, Catalonia, the Basque Country, and Madrid—groups have been investigated that share images and videos of extreme explicit violence, pornography, paedophilia, animal abuse, sexism, homophobia, and fascism ([39]; [46]).

The mechanism for including so many children is to make the minors themselves administrators of the group, allowing them to bring in their peers ([46]). The supposed objective of these mass chats is to make the group as large as possible, with names such as “add people until we become famous”, “add people until we reach a million”, or “add everyone you know on WhatsApp” ([46]). Generally, the creators of these groups are often young people from the school environment itself ([46]). For this reason, minors are often identified as both victims and perpetrators ([46]). However, the important thing is not who created the group, but who disseminates the content, because sometimes it is created with one intention and then transformed ([46]).

Most of the minors who were included, without their consent, in these mass chats with violent content showed boredom and disgust, but none of them left ([46]). Two hypotheses are being considered to understand this permanence.

The first has to do with minors not being aware that sharing certain content through digital platforms is a crime, depending on the content and the victim, crimes of humiliation, crimes against privacy, child pornography, corruption of minors, etc. There is a common misconception that certain laws do not apply or operate in the digital world ([10]; [46]). It is believed that what happens online is less serious. However, it has been shown that virtual gender-based violence has serious effects on people’s emotional well-being in their offline lives ([53]).

The second hypothesis pertains to the sense of belonging to a group, which is particularly important during puberty and adolescence ([51]). Participating in these types of chats can promote a sense of belonging or affiliation to a tribe ([3]; [51]), but it also increases vulnerability or puts the well-being of minors at risk if these spaces are permeated by dynamics of abuse, exclusion, or violence ([51]).

Concern about these groups arises because early exposure to sexual violence and pornographic content encourages the normalisation of such behaviour and increases the risk of future reproduction of sexist practices ([13]; [21]; [35]). Therefore, when the existence of these chats is detected in any context, whether in a family or school setting, it would be advisable to inform the rest of those involved, as has happened in some cases where schools have taken responsibility for communicating this to families ([46]). In addition, as a preventive measure, it would be beneficial to incorporate effective sexual education more forcefully into basic education curricula, as this is an effective strategy for preventing the normalisation and emergence of sexist behaviour both offline and online ([4]; [7]). Likewise, it would be advisable to educate minors about the existence of digital rights ([19]), which would help minimise the trivialisation of certain violent behaviours and foster a sense of accountability on digital networks and platforms.

### 6.4. Incitement to Sexist Violence Through Video Games

The development and distribution of video games with content that directly incites sexual violence against women is one of the most explicit forms of sexist violence in the digital world. The video game “No Mercy” is a clear example of this.

“No Mercy” has been described as a “non-consensual sex” simulator, where players are encouraged to become “women’s worst nightmare”, embodying the type of man who “never takes no for an answer” ([5]; [16]). The game allowed players to control a character who rapes, tortures, and kills women ([49]).

As a result of social pressure, and specifically from feminist associations, “No Mercy” has been removed from the Steam platform—the world’s largest video game distributor—in several countries: Australia, Canada, Spain, the United States, and the United Kingdom ([16]; [49]).

As mentioned above, there is scientific evidence linking the consumption or perpetration of acts of male sexual violence in digital environments with real crimes ([13]; [21]; [35]). These types of video games fuel and perpetuate a culture of misogyny and rape by dehumanising women, objectifying them, and establishing a relationship of sexual power over them ([5]).

Fortunately, there are now “Guidelines on prohibited artificial intelligence practices established” ([19]) which, although they do not specifically mention anything related to video game content, it opens the door or reinforces the possibility of sanctioning agencies or individuals responsible for distributing digital content that incites sexual abuse or violence through online platforms or games ([60]).

However, the answer to ending this social scourge in the offline and online world is not punishment or prohibition, but education ([8]). Hence, there is a demand to explicitly and categorically incorporate effective sexual education into the curricula of all educational stages, as it is an effective alternative to curb the normalisation and adoption of sexist behaviour in both face-to-face and digital contexts ([4]; [7]). Furthermore, taking into account the differences in the use of ICTs across genders reflected in the cases described and in the scientific literature ([53]), it is advisable to encourage empathy, moral commitment and the internalisation of norms in male minors ([12]; [15]), and in female minors, facilitate empowerment through sisterhood ([32]).

## 7. Measures/Actions to Minimise the Digital Gender Divide: Let Us Fight for Girls’ Well-Being, So That They Are Not “Taken Advantage of”

Given the context described above, in which new forms of digital gender-based violence are flourishing and running rampant, institutional responsibilities are crucial. The digital age demands a collaborative effort and the transfer of “anthropo-ethics” to the digital world, to understand that the individual, the social, and the global—the consideration of the human species as a whole—are interconnected ([42]). It involves the regeneration of democracy, civility, solidarity, and responsibility ([42]). Taking advantage of new technologies without forgetting ethics enables individuals to develop as role models in life, both offline and online ([54]), as it would be unthinkable to exclude, benefit, or ensure the well-being of a few while harming the most vulnerable.

It is essential to create a safe and beneficial digital environment, particularly for children and adolescents. The approach must be intersectional and multifaceted (policies, formal training, and community families). Only in this way can effective strategies be developed to ensure safe, inclusive, and equitable digital environments, in which no form of violence is encouraged, and the well-being of children and young people is not at risk.

### 7.1. European Policies or Plans Aimed at Guaranteeing Digital Rights and Security

Digitalisation must integrate the protection of citizens’ rights—especially those of minors—and not be limited merely to technological advances ([24]). Several actions can help us exemplify this necessary shift.

The actions to be presented are part of the 2030 policy programme ‘Path to the Digital Decade’ ([11]). This policy programme, which has been in development for several years, was approved in 2022. The objectives are to ensure digital skills among the population (basic skills for citizens and specialised skills for ICT professionals); secure and sustainable digital infrastructures through improved connectivity, minimising any type of risk as much as possible; digital transformation of businesses; and digitisation of public services (see Figure 2).

Within this framework, the European declaration on digital rights and principles for the digital decade ([18]) (see Figure 3) has been published, preceded by the Charter of Digital Rights published by the European Commission in 2021 to equalise the protection of individuals in the offline and online worlds ([24]).

Also published in 2022 is A digital decade for children and youth: the new European strategy for a better internet for kids (BIK+) ([17]), which is based on three pillars: (1) safe digital experiences, based on protecting children from harmful and illegal online content, conduct, and risks and improving their well-being through a safe, age-appropriate digital environment; (2) digital empowerment so that children acquire the necessary skills and competences to make informed choices and express themselves in the online environment safely and responsibly; and (3) active participation, respecting children by giving them a say in the digital environment, with more child-led activities to foster innovative and creative safe digital experiences.

In February 2025, guidelines on prohibited artificial intelligence practices, established by Regulation (EU) 2024/1689 (AI Act) ([19]), were released. Although the guidelines are not legally binding, they provide guidance for supervision, compliance assessment, and the development of operational policies. In addition, specific AI practices are outlined that are expressly prohibited because their potential to cause harm is considered high and incompatible with fundamental European values. This specificity facilitates the transfer of risks to different situations. It ensures the following requirements or principles: proportionality, limitation of data use, transparency, human oversight, accountability, security, respect for the autonomy, dignity, and integrity of individuals, and non-discrimination.

However, these European frameworks are not sufficient to ensure the protection of minors in the digital world. Laws would be needed to explicitly regulate institutions such as families, educational centres, sports centres, healthcare facilities, and other similar entities. For example, schools should be encouraged to include content on digital rights and cybercrime in the curriculum, as well as promoting the use and quality of ICTs (critical, ethical, responsible use); to define protocols for action to intervene in cases of possible online violence occurring among students (cyberbullying, sextortion, grooming, dissemination of images without consent); and to ensure that the organisational chart includes people responsible for child and youth well-being and/or people in charge of developing digital citizenship skills (DigComp), teaching skills (DigCompEdu) and organisational skills (DigCompOrg).

Mere guidelines do not directly inspire anyone to change or transformation. The realities reviewed call for direct, concrete, and forceful action.

### 7.2. Contexts of Critical Digital Literacy in Formal Education: Schools and Universities

To understand the following measures that will be proposed, it is necessary to avoid “technosolutionism,” which tends to delegate the generation of responses to social, political, educational, or cultural problems to digital technology ([43]). Such proposals are often simplistic, partial, and/or biased, as they overlook the complexity of the issues by ignoring structural factors and the intersectionality that surrounds them.

Therefore, redefining the algorithms that govern social media is a partial and ineffective measure, as it is insufficient on its own. Citizens need to be empowered and develop complex and critical thinking skills to inhabit and live in the digital world in a safe, inclusive, and not only fair but equitable manner, recognising the structural inequalities that affect girls and adolescents ([42]; [44]; [57]; [45]; [54]; [63]). And for this, there is no better context than formal education ([8]; [44]; [57]).

#### 7.2.1. Critical Digital Literacy in Schools

These new forms of virtual gender-based violence make it more important than ever to equip citizens—especially children and adolescents—with digital skills. Of the five competence areas of DigComp 2.2 ([63]), the most decisive in tackling this social cyber scourge would be the fourth, relating to digital safety, because it encompasses not only the protection of one’s own devices and personal data (privacy control, password management) and those of others, but also health and psychological, emotional and relational health and well-being in digital environments, enabling minors to identify risky consumption or interactions (harmful content on social networks, instant messaging chats or video games; sharing personal/private data belonging to themselves or others without consent; sexting; creation and dissemination of deepfakes; and cyberbullying, sextortion, grooming, etc.).

Furthermore, this area of competence facilitates understanding of the ethical and legal implications of using technology. In this regard, teaching minors that there are digital rights that protect us and must be respected helps them understand and internalise that certain actions, even if carried out in the virtual environment, constitute crimes ([44]; [57]). It is necessary to make minors aware that virtual gender-based violence has serious effects on the emotional well-being of people in their offline lives ([53]). Accountability urges minors not to engage in gender-based violence in any digital environment and to report it when they are victims or witnesses of it. This will encourage minors to refrain from feeding their sense of belonging or affiliation to a group through participation in digital groups permeated by dynamics of abuse, exclusion, or violence ([3]; [15]; [51]).

Competence in safety contributes significantly to the development of critical digital literacy, which, to be truly effective, must be adapted to personal characteristics, such as gender, i.e., differences in the use of ICTs across genders. In relation to male minors, it is necessary to reinforce effective sexual education from a feminist and intersectional approach, raise awareness of gender equality as a right, promote empathy (both cognitive and affective), moral commitment, the internalisation of norms, and the responsible use of ICTs ([4]; [7]; [9]; [12]; [15]; [22]). On the other hand, in the case of female minors, their empowerment should be achieved through sisterhood ([32]), that is, by building relationships based on solidarity, alliance and mutual support among women, which, based on the recognition of structural inequalities derived from patriarchy, are based on promoting equality and eradicating gender-based violence in all its forms, including digital violence. This would involve appealing for social transformation through female and feminist care networks, whether digital or not, whose values are trust, respect, and co-operation.

Finally, given that there is scientific evidence linking the consumption or perpetration of acts of gender-based sexual violence in digital environments with real crimes ([13]; [21]; [35]), institutions such as schools cannot stand on the sidelines and refuse to educate minors, because this is currently one of the most effective measures for ensuring the well-being of children and young people in the digital world and reducing the digital gender divide ([44]).

#### 7.2.2. Critical Digital Literacy for Future Teachers in Education Faculties

It is undeniable that Digital Competence for Educators (DCE) is key for children and adolescents to develop DC in educational centres ([37]). DCE was defined as the set of knowledge, skills, and attitudes that teachers need to master ICT and integrate it in a transformative way into the teaching–learning processes implemented during daily practice ([55]; [37], [36]). Currently, DCE refers to “the integration of knowledge, skills, abilities, and attitudes that must be simultaneously brought into play to perform their duties by implementing digital technologies and to solve problems and unforeseen events that may arise in a specific situation as education professionals” ([50]). DCE is independent of the educational stage, subject, or type of teaching ([50]). It is therefore general in nature, and DCE is structured around the functions shared by all teachers ([50]).

The Digital Competence Framework for Educators, known as DigCompEdu, establishes six areas of competence ([50]): (1) professional engagement, (2) digital resources, (3) teaching and learning, (4) assessment, (5) empowering learners, and (6) facilitating learners’ digital competence. Like DigComp, each of the six areas of DigCompEdu consists of 22 specific competences (see Figure 4). Six levels of proficiency are established. In the first two stages, Newcomer (A1) and Explorer (A2), educators assimilate new information and develop basic digital practices; in the following two stages, Integrator (B1) and Expert (B2), they apply, further expand, and structure on their digital practices; and at the highest stages, Leader (C1) and Pioneer (C2), they pass on their knowledge, critique existing practice, and develop new practices.

The six areas are organised into three blocks ([50]): educators’ professional competences (made up of competence area 1), educators’ pedagogical competences (made up of competence areas 2, 3, 4, and 5), and learners’ competences (made up of competence area 6) (see Figure 4). The fundamental basis of DigCompEdu is supported by competency areas 2, 3, 4, and 5, which constitute the so-called pedagogical competencies essential for teachers to integrate ICT into teaching and learning processes inclusively and innovatively ([50]).

Considering the issue at hand, new forms of virtual gender-based violence, it is worth noting that competence area 1, educators’ pedagogical competences, is particularly important, as it is where institutional protocols for cyber coexistence and cyberbullying prevention would be located from an educational, inclusive, gender-equitable, and restorative perspective. In this regard, recent studies emphasise that digital training for teachers and the existence of clear and up-to-date protocols are decisive factors in ensuring the protection of students and promoting an ethical and respectful digital school culture ([22]).

### 7.3. Contexts of Critical Digital Literacy in Informal Education: Parents, Carers, and Guardians

In the face of new forms of gender-based violence in digital environments, it is not only schools as educational institutions that must take an active role; parents, carers, and guardians must also take responsibility for the digital and emotional education of their children. Although these may be conditioned by various divides—access, use, and exploitation—it is imperative that they contribute to the critical digital literacy of children and adolescents to the extent possible. Delegating this task is no longer an option.

This imperative places pressure on the institutions responsible for promoting digital literacy among parents or caregivers. As a result, many countries are implementing family digital education programmes ([44]).

Digital parenting must evolve in tandem with the child’s age ([44]). In childhood, firm rules are needed to limit exposure time and close supervision of the digital content consumed. In adolescence, rules should be more flexible and agreed upon, with opportunities for dialogue based on trust and unconditional acceptance.

Furthermore, it is essential to be healthy role models and avoid engaging in “distracted parenting” due to screen dependency ([44]). To this end, parental self-assessment and self-regulation are key in terms of the time spent online and the content consumed. The digital world should not hinder the creation and strengthening of emotional bonds and communication with children ([44]).

Active and responsible digital parenting is essential, involving a shared responsibility for ensuring their children’s digital literacy. For a safe digital world, parents and educators must be part of a trained network, so that the home can serve as a refuge for growth, considering the coexistence of both the offline and online worlds.

## 8. Discussion and Conclusions: From Institutional Security to Personal Digital Resilience

The results of the documentary analysis conducted reveal several phenomena that substantiate the need to reconceptualise the digital gender divide (see Figure 5). Beyond disparities in access and use, the findings highlight that the dimension of empowerment—understood as the ability to benefit safely, meaningfully and ethically from digital participation, without limiting this dimension to agency and autonomy—is profoundly affected by gender inequalities. Within this dimension, there is clear evidence of the digital sexual exploitation of girls and young women, manifested through practices such as grooming, sextortion, the production of deepfakes, and the circulation of violent and pornographic content in online spaces such as mass chats or video games.

Cases such as those of Amanda Todd, Molly Russell, the Almendralejo deepfakes, WhatsApp pornographic mass chats and the *No Mercy* video game underscore the urgent need for updated legal frameworks addressing digital sexual exploitation and the incitement of gender-based hatred in online environments. They also highlight the importance of prevention strategies that combine critical digital education with affective-sexual training.

In legal terms, the publication of the *Guidelines on Prohibited Artificial Intelligence Practices established by Regulation (EU) 2024/1689 (AI Act)* ([19]) represents a relevant step, as it provides non-binding but essential guidance for oversight, compliance assessment, and policy development.

From an educational perspective, it is necessary to foster complex and critical thinking among citizens to ensure a safe and inclusive digital environment—one that promotes not only justice but also equity by acknowledging the structural inequalities affecting girls and adolescents ([7]; [44]; [57]). Formal education remains the most effective space to achieve this goal ([7]; [8]; [44]; [57]). Among the initiatives to be followed, the programme proposed by [7] ([7]) stands out for providing learners with solid knowledge about sexuality, consent, and healthy online relationships to counter gender-based violence in digital contexts.

Believing that a totally secure digital world is currently possible is a naive and unattainable utopia. It would be like thinking that a real world without risks exists.

And to think that the best way to prevent digital dangers is to ban people from inhabiting the online world would be similar to thinking that the answer to ensuring the safety and well-being of children and adolescents is to keep them from leaving the house.

It is extremely important to have institutional measures in place to protect against digital threats. However, they must be prepared for when all this fails, because it inevitably will.

In these situations, what will save minors is critical digital literacy, as it addresses the need to develop personal digital resilience—the ability of children and adolescents to overcome negative online experiences and transform vulnerability into strength by learning from past experiences ([17]). This framework calls not only for children to be protected but also for them to be digitally competent and resilient.

Finally, it is necessary to recognise that this study presents some limitations that should be acknowledged, but which do not minimise the importance of the results. First, the selection of cases was based primarily on their visibility in *Google News*, which, while useful for identifying highly mediatised events, may not capture the full scope of digital gender-based violence affecting minors across different regions or platforms. Future research could broaden this approach by analysing the impact of these cases across various social, institutional, specially educational, and academic contexts, to better understand their societal resonance. Finally, expanding the corpus to include emerging cases or new digital phenomena would allow for a more dynamic and comprehensive understanding of how the digital gender divide evolves and manifests through diverse forms of online violence.

## Figures and Tables

**Figure 1 behavsci-15-01568-f001:**
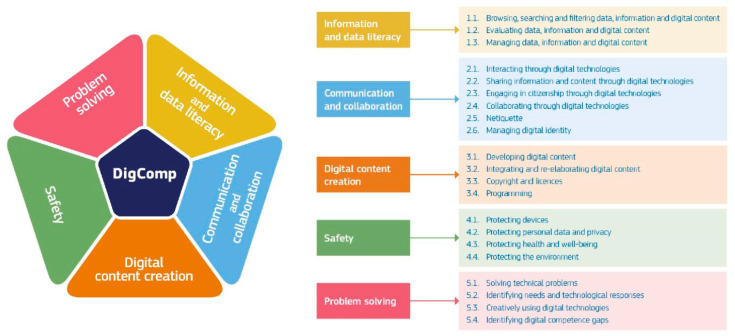
Digital competence areas and specific digital competences of DigComp. *Source*: https://joint-research-centre.ec.europa.eu/projects-and-activities/education-and-training/digital-transformation-education/digital-competence-framework-citizens-digcomp_en (accessed on 10 October 2025).

**Figure 2 behavsci-15-01568-f002:**
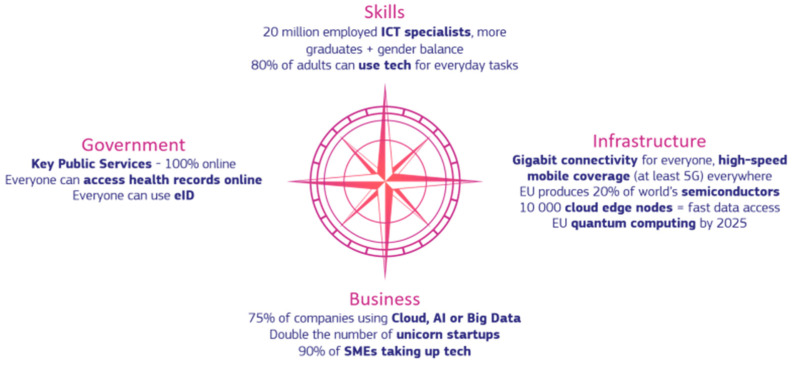
Four cardinal points of the Digital Decade. *Source*: https://digital-strategy.ec.europa.eu/sites/default/files/styles/extra_large/public/2023-01/digital%20decade%20compass_updated%20January.png?itok=PKmHJ_wp (accessed on 10 October 2025).

**Figure 3 behavsci-15-01568-f003:**
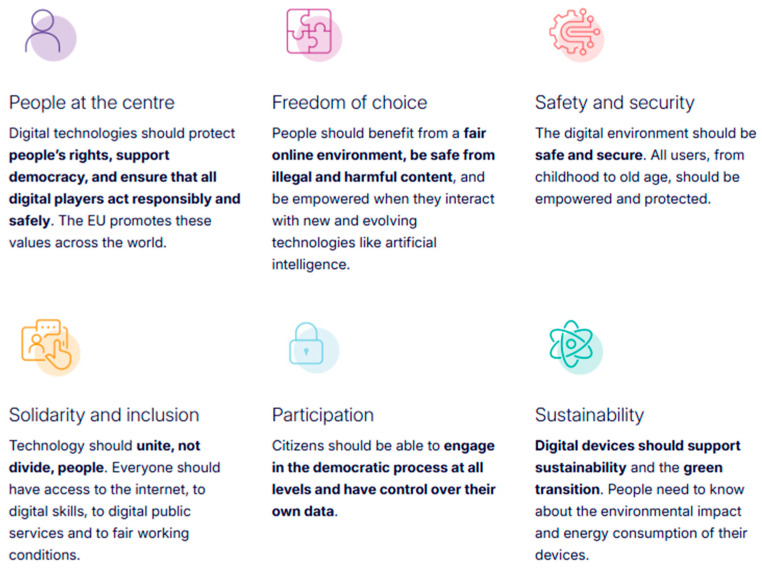
Digital rights. *Source*: https://commission.europa.eu/strategy-and-policy/priorities-2019-2024/europe-fit-digital-age/europes-digital-decade-digital-targets-2030_en (accessed on 10 October 2025).

**Figure 4 behavsci-15-01568-f004:**
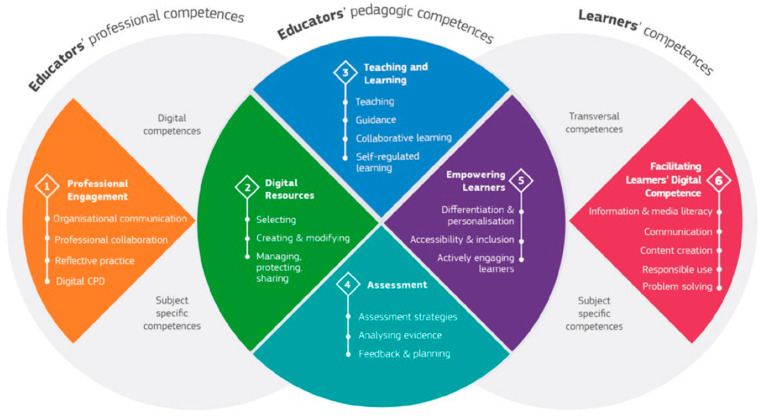
The synthesis of the DigCompEdu framework. *Source*: ([50]).

**Figure 5 behavsci-15-01568-f005:**
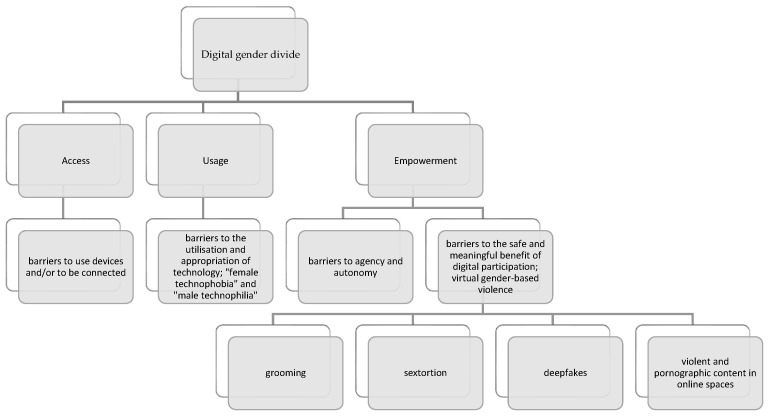
Reconceptualising the gender digital divide. *Note*: Own elaboration.

**Table 1 behavsci-15-01568-t001:** Manifestations of digital gender-based violence.

Type of Digital Gender-Based Violence	Definition
Exposure to harmful content	Access to communities or algorithms that promote self-harm, anorexia, violence, or suicide.
Hate speech	Spreading discriminatory or hateful messages (racism, sexism, homophobia, etc.) in digital environments.
Sexist violence in immersive environments	Harassment, insults, simulated assaults, or incitement to sexual violence within online games.
Sexting	Sending sexual photos, videos, or messages via digital devices. Although it may be consensual, it carries the risk of non-consensual dissemination.
Sexual deepfakes	Creation of fake images or videos using Artificial Intelligence (AI) to damage reputation or harass.
Grooming	A strategy of sexual manipulation used by an adult towards a minor on the internet, seeking to obtain intimate material or encounters.
Sextortion	Blackmailing the victim with the threat of disseminating intimate material if they do not comply with demands (money, more images, sexual favours).
Revenge porn	Publishing or sending sexual content without the consent of the person involved.

*Note*: Own elaboration.

**Table 2 behavsci-15-01568-t002:** Categorization of digital gender-based violence.

Category	Subcategory (Due to)	Operational Definition	Case (Year of Occurrence)
Self-harm and suicide among minors	Grooming, sexting or sextortion (Sui_Groom-Sext)	Cases in which sexual harassment, manipulation, or non-consensual dissemination of intimate images trigger situations of humiliation, anxiety, isolation, or suicide.	Amanda Todd (2012) International child exploitation network known as “764”
	Exposure to harmful content	Cases in which interaction with social media exposes minors to harmful digital content that directly affects their emotional well-being and leads to self-destructive behaviours.	Blue Whale Challenge (2015) Molly Russell (2017) Momo Challenge (2018)
Digital sexual exploitation of underage girls through deepfakes		Cases in which artificial intelligence tools are used to generate or manipulate fake sexual images of minors for purposes of exploitation, harassment, or blackmail.	Fake nudes of Almendralejo (2023), of Cyprus (2025), and of Limassol (2025)
Consumption of violent and hateful content in mass chats		Cases in which minors are exposed to or participate in large-scale online chat groups where violent, pornographic, or hate-promoting material circulates, reinforcing sexist and aggressive behaviours.	WhatsApp pornographic chat in Spain (2023)
Incitement to sexist sexual violence through video games		Cases in which gaming platforms or digital communities reproduce and normalise sexual violence against women and girls through interactive dynamics, language, or visual content.	“No Mercy” video game (2025)

*Note*: Own elaboration.

## Data Availability

The original contributions presented in this study are included in the article. Further inquiries can be directed to the corresponding author.

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
