# Peer review of "Reconceptualising the Digital Gender Divide, Accommodating New Forms of Virtual Gender-Based Violence"

_behavsci, 2025, doi:10.3390/bs15111568_

Round 1
Reviewer 1 Report
Comments and Suggestions for Authors
Dear Authors,
The article addresses a current and relevant topic: the intersection between the digital gender gap and new forms of gender-based violence mediated by digital technologies. The article offers contributions to policy-making and digital education with a gender perspective. The article is well structured, with a robust and updated theoretical foundation and practical examples that materialize the theme are presented. The idea of "personal digital resilience" represents an interesting and well-founded conceptual contribution.
Since this is a review/essay article, there is no formal empirical methodology. Still, the method of selecting cases and references could be referenced.
Despite the quality of the article, the following improvements are suggested:
- Include a brief methodological subsection referring to how the bibliographic sources were selected (criteria and time period). And, also mention the selection criteria for the cases presented.
-Include in the "Conclusion" section a brief summary of the theoretical and practical implications of the work presented in the article.
Recommendation: Minor revision
Author Response
I have attached a PDF file explaining the changes.

Reviewer 2 Report
Comments and Suggestions for Authors
Reconceptualising the Digital Gender Gap, accommodating new forms of virtual gender-based violence
General overview
The manuscript presents a critical and updated review of the concept of the digital gender disparity by linking it to the new forms of virtual gender-based violence that particularly affect both girls and adolescents. It embraces an intersectional feminist approach and proposes several aspects such as critical digital literacy measures, European policies for the protection of digital rights, and strengthening personal resilience.
The text is well-structured and consistent, while using up-to-date sources (2022-2025). It is a relevant contribution to the field of gender studies, digital literacy, and psychosocial health within virtual environments.
Strengths
- Current topics: The manuscript analyzes recent cases of digital exploitation, fake images or videos generated by artificial intelligence (deepfakes), online sexual harassment and manipulation of minors (grooming), and sexist video games, while linking empirical evidence with current European policy frameworks (e.g., AI Act 2024/1689 and Path to the Digital Decade).
- Conceptual rigor: The manuscript contains a solid review of the DigComp 2.2 and DigCompEdu frameworks, which is related to digital competence, well-being and technological ethics.
- Human rights and gender approach: the manuscript deals with feminist and childhood perspectives, supported by organizations such as UNESCO, OECD, and UNICEF.
Aspects that need improving
Formal Structure:
- Although the manuscript is comprehensive, it is necessary to summarize or integrate sections 4.1–4.4 (case studies) to avoid descriptive redundancies.
- I strongly recommend that authors should include a comparative summary table between the different types of digital violence reviewed (self-harm, deepfakes, mass videos that send inappropriate content (mass chats), video games).
Methods
- The search, inclusion, or exclusion criteria for literature are not specified. A brief methodological description is necessary to increase the study's reproducibility.
Discussion
- It is necessary for authors to deepen the relationship between critical digital literacy and psychosocial resilience (for example, studies on digital resilience in adolescents).
Conclusions:
- The conclusions are very robust. However, it would be useful if the authors included a section concerning operational recommendations for public policies and teacher training.
Useful proposals for elaborating public policies and inclusive digital literacy programs.
Author Response

(The authors gave the same response as above.)

Reviewer 3 Report
Comments and Suggestions for Authors
The article presents a topic of great personal interest to me, and I really liked the approach and the care taken to include sections focused on family, schools, teacher training, etc.
I have a few suggestions to improve the methodological robustness and scientific clarity of the article.
The article begins by focusing on Digicomp. Before that, I suggest a short introduction to present the reader with what follows. If we start by giving isolated concepts, it will not hold the reader's attention.
One concern I have is that the article is in the reviews category, but does not have a methodology section. How were the sources selected? What were the inclusion and exclusion criteria? If the intention is not to be a review, but rather something like an essay, please make this clear in the introductory paragraph I suggested earlier.
I understand why you used so many journalistic sources, but I believe you should justify this fact (in a methodology section).
Upon reading the text, it appears to me that there is a lack of interconnection between the sections and a logical structure to the argument. Currently, they appear to be separate topics rather than distinct parts of a single text. I suggest better integration, perhaps with a concluding paragraph that explains the connection to the following section. Additionally, in some passages, I noticed a lack of citations to support the argument, which made it read more like an opinion.
In the title, the author claims they will reconceptualize, but in the text, they ultimately fail to focus on this reconceptualization. I suggest inserting a conceptual diagram that integrates access, use, and enjoyment with the axes of violence and intersectionality, and presents the operationalization of the constructor.
I felt that an explicit section on the study's limitations and future directions was missing.
Regarding references, as the article is in English, the APA standard should be followed when the source is in a language other than that of the article, with the English translation in brackets, for example:
Cunha, M. N. (2024). Empatia: A chave para combater o bullying entre crianças e adolescentes [Empathy: The key to combating bullying among children and adolescents]. RCMOS - Revista Científica Multidisciplinar O Saber, 1(2), 1–13. https://doi.org/10.51473/rcmos.v1i2.2024.794
Author Response

(The authors gave the same response as above.)

Round 2
Reviewer 3 Report
Comments and Suggestions for Authors
Dear Author,
Thank you for carefully addressing my suggestions. The manuscript has improved significantly, and it is clear that substantial effort was dedicated to the revision. I recommend the paper for publication in its present form.